# An all-in-one tetrazine reagent for cysteine-selective labeling and bioorthogonal activable prodrug construction

Xinyu He[1], Jie Li[1], Xinxin Liang[1], Wuyu Mao[1], Xinglong Deng[2], Meng Qin[3], Hao Su[4] & Haoxing Wu [1,2] ✉

The prodrug design strategy offers a potent solution for improving therapeutic index and expanding drug targets. However, current prodrug activation designs are mainly responsive to endogenous stimuli, resulting in unintended drug release and systemic toxicity. In this study, we introduce 3-vinyl-6-oxymethyl-tetrazine (voTz) as an all-in-one reagent for modular preparation of tetrazine-caged prodrugs and chemoselective labeling peptides to produce bioorthogonal activable peptide-prodrug conjugates. These stable prodrugs can selectively bind to target cells, facilitating cellular uptake. Subsequent bioorthogonal cleavage reactions trigger prodrug activation, significantly boosting potency against tumor cells while maintaining exceptional off-target safety for normal cells. In vivo studies demonstrate the therapeutic efficacy and safety of this prodrug design approach. Given the broad applicability of functional groups and labeling versatility with voTz, we foresee that this strategy will offer a versatile solution to enhance the therapeutic range of cytotoxic agents and facilitate the development of bioorthogonal activatable biopharmaceuticals and biomaterials.

Enhancing the efficacy of cancer treatment relies on increasing the effective concentration of chemotherapy drugs at the target site while minimizing toxicity to healthy tissues[1,2]. Consequently, several prodrug design strategies have emerged, including the conjugation of cytotoxic drugs to appropriate antibodies, peptides, or polymers[1,3–6]. Among these strategies, peptide-drug conjugates (PDCs) offer the most well-defined chemical structures, high binding affinities, and tunable pharmacokinetics, positioning them as the next generation of precision therapeutics[5,7,8]. With the recent clinical approval of two PDCs[9,10], the development of new PDC design strategies has garnered significant attention[6,11].

PDCs consist of a chemotherapeutics agent covalently linked to a peptide as a prodrug, that binds to a receptor molecule that is specifically expressed within a target tissue. Ideally, PDCs should remain stable in the bloodstream, but upon reaching the disease site, the linkage should be selectively cleaved, thereby releasing the cytotoxic payload. To fulfill this requirement, researchers have developed various cleavable linkers that respond to specific stimuli such as pH, reducing potential, or enzymes that are overexpressed in the tumor microenvironment[5,12]. Unfortunately, these endogenous stimuli are not completely absent in normal tissues, leading to unintended exposure of healthy tissues to the drug. To overcome this nonspecific toxicity, the next-generation PDCs will include novel linker chemistries that are orthogonal to the native biological milieu.

Bioorthogonal cleavage reactions[13–16], which are inert within the living system but triggered only by the corresponding bioorthogonal

[1]Department of Radiology and Huaxi MR Research Center (HMRRC), Functional and Molecular Imaging Key Laboratory of Sichuan Province and Frontiers Science Center for Disease Related Molecular Network, West China Hospital, Sichuan University, Chengdu, China. [2]Key Laboratory of Drug-Targeting and Drug Delivery System of the Education Ministry and Sichuan Province, Sichuan University, Chengdu, China. [3]National Chengdu Center for Safety Evaluation of Drugs, State Key Laboratory of Biotherapy, West China Hospital, Sichuan University, Chengdu, China. [4]College of Polymer Science and Engineering, State Key Laboratory of Polymer Materials Engineering, Sichuan University, Chengdu, China. ✉e-mail: haoxingwu@scu.edu.cn

partner, allow for specific bond breakage and release of payloads from the delivery machinery[17–26]. In order for drugs to be released from the peptide in vivo, the cleavage reaction must be sufficiently rapid to occur at the drug's local concentration, typically in the submicromolar range[17,27], and to compete with the relatively fast peptide clearance rate.

Recently, Robillard and co-workers and our group sequentially reported a new generation bioorthogonal "click-to-release" approach in which amines[21], phenols, or carboxylic acids[22] are liberated from a methylene tetrazine moiety. The second-order rate constant for this cleavage strategy is orders of magnitude greater than those observed in previously reported strategies which release payload from allylic-substituted *trans*-cyclooctenes[21], allowing prodrug activation to occur at nanomolar concentrations[27]. We envision that this new generation cleavage reaction could be employed in the construction of next-generation PDCs, enabling the release of drugs through spatio-temporally regulated tetrazine bioorthogonal reactions.

Another critical aspect in the design of PDCs is the introduction of prodrugs that do not disturb the architecture of the peptide and its binding affinity for the target[7]. Therefore, PDCs must be constructed with linkages of minimal size. In addittion, since peptide chains contain multiple reactive functional groups, the development of a chemoselective modification strategy that can be performed on native peptides under mild conditions would facilitate the modular discovery of PDCs by greatly reducing tedious synthetic steps[28].

To optimize the PDC method, this study aims to devise an all-in-one linker design strategy for introducing minimal bioorthogonal tags and drugs onto peptides in the late stage in a modular and chemoselective manner, resulting in the development of bioorthogonal cleavable PDCs. As tetrazine is the most electron-poorest ring system[29], we considered that a vinyl group adjacent to the tetrazine ring would be sufficiently electron-deficient and highly reactive to react chemoselectively with the Cys thiol group under mild conditions. Therefore, we envisioned 3-vinyl-6-oxymethyl-tetrazine (voTz, Fig. 1) as a reagent for PDC design. The oxymethyl moiety was expected to be linked with drug functionalities, resulting in bioorthogonal cleavable prodrugs. The reactive vinyl group would enable unprotected peptide bioconjugation. Importantly, we hypothesized that the resulting dialkyl-tetrazine would exhibit sufficient stability in the circulation and excellent reactivity towards dienophiles for bioorthogonal release on demand. In addition, we used this reaction to produce a probe in which fluorescence is activated by the tetrazine bioorthogonal reaction. This turn-on feature allowed us to directly visualize the dynamics of drug release using quantitatively fluorescent signals. Given its attractive properties, we anticipate that this all-in-one linker will be used to modularly construct various bioorthogonal cleavable PDCs and study their therapeutic effects.

## Results

### Chemoselective labeling with vinyltetrazine
We initiated our study by synthesizing phenyl vinyltetrazine (1) as a model compound to assess the chemoselective labeling of Cys with vinyltetrazine. To our satisfaction, when a stoichiometric amount of 1 was mixed with Cys in phosphate buffer (PB, pH 8.0) with 30% MeCN as co-solvent in an open-air environment, we successfully detected the desired labeling product Cys-1 within 30 min, with quantitative transformation and 97% yield (Fig. 2A). We confirmed and further characterized the product using LC−MS (Supplementary Fig. 1) and operando NMR spectroscopy (Supplementary Fig. 11). In addition, a second-order rate constant of $13 \pm 0.8\,M^{-1}\,s^{-1}$ was determined for the labeling reaction between 1 and Cys (Supplementary Fig. 12). To investigate the chemoselectivity of the labeling reaction, we conducted a competition experiment. In the presence of a large excess of amino acids with chemically active side chains (10 equiv. each of Lys, Ser, Tyr, and Arg), we detected only the formation of the desired Cys-

labeled product (Fig. 2B and Supplementary Fig. 13). This observation demonstrates the excellent chemo-selectivity of vinyltetrazine toward Cys. Additionally, the labeling reaction can be facilely performed under a wide range of conditions: we achieved similarly high labeling yields in the presence of a variety of buffer combinations at neutral pH and aqueous or organic solvents, and when including the peptide and labeling reagent at wide concentration ranges (Supplementary Table 1 and Supplementary Figs. 2–10).

### Design and modular preparation of peptide conjugates using the voTz linkage
Our next questions revolved around the possibility of developing modular voTz-caged drugs or molecular probes, and whether these caged molecules could be conjugated to free peptides at the late stage. To address these questions, we designed a key precursor, hydroxymethyl-tetrazine (2), which was prepared in three steps from commercially available materials on a multigram scale with an overall yield of 20% (see details in the Supplementary Information). Starting with compound 2, we conducted NBS bromination and alkylation to obtain three phenol ether intermediates (3a–5a) in a straightforward manner. After removing the TBS group, a cascade mesylation and elimination produced the desired voTz-caged ethers (3–5) in moderate to good yields (Fig. 2C, 47–79%). The versatile precursor 2 could be converted to carbamate 6a in a 91% yield by treating it with isocyanate. Additionally, it could be transformed into an active carbonate (7a) with a 90% yield. Following the same deprotection and cascade mesylation and elimination procedure, we successfully obtained the voTz-caged carbamate 6 and voTz-caged carbonate ester 7b in moderate yields (Fig. 2C, 50–51%). With an aminolysis step involving the reaction of 7b with doxorubicin, we formed the desired carbamate 7 in good yield. Consequently, we obtained five voTz derivatives, including two model compounds (3 and 6), caged fluorophore 4, and two prodrugs containing a camptothecin analog (SN-38, 5) or doxorubicin (Dox, 7).

To test the labeling at the late stage, we chose nine peptides, which comprised five model peptides ($P_1$–$P_5$) consisting of 4–17 amino acids and containing all common nucleophilic amino acid residues. The other four were tumor-targeting peptides: $P_6$ (FSH1), targeting follicle-stimulating hormone receptor[30]; $P_7$ (BP9a), targeting human transferrin receptors[31]; $P_8$ (RGD) targeting integrin receptors on the tumor vasculature[32]; and $P_9$ (peptide 18–4), which targets keratin 1 in breast cancer cells[33,34]. Notably, the Cys residues in these peptides were situated in different sequence environments.

We rapidly conjugated stoichiometric amount of voTz derivatives (1.1 eq.) into $P_1$–$P_9$, resulting in single-labeled products ($P_1$–4–$P_9$–7) with yields of 88–97%, under the abovementioned buffer conditions within 2 h in PB, pH 8.0 with 30% MeCN as co-solvent. The conjugation can also be smoothly performed at lower concentrations in the same condition with a yield of 89%, or high concentrations in organic solvent for scale-up with the same high yields (91–94%). To our delight, both the ether and carbamate bonds remained stable during labeling, and no cleavage by-products were formed according to LC−MS analyses. Consequently, we obtained 20 peptide conjugates, including two peptide-fluorophore conjugates and four peptide-drug conjugates containing SN-38 or Dox prodrugs (Fig. 2D and Supplementary Fig. 14–34).

### Bioorthogonal cleavage of PDCs and nano-structure characterization
Having developed voTz as a linker reagent to construct peptide conjugates, we next aimed to evaluate the physiochemical properties and applicability of these conjugates to the bioorthogonal click-to-release strategy (Fig. 3A). Considering that the resulting peptide conjugates contain a relatively hydrophilic peptide and a hydrophobic tetrazine caged prodrug (SN-38 or Dox) or fluorophore (FL), we predicted that

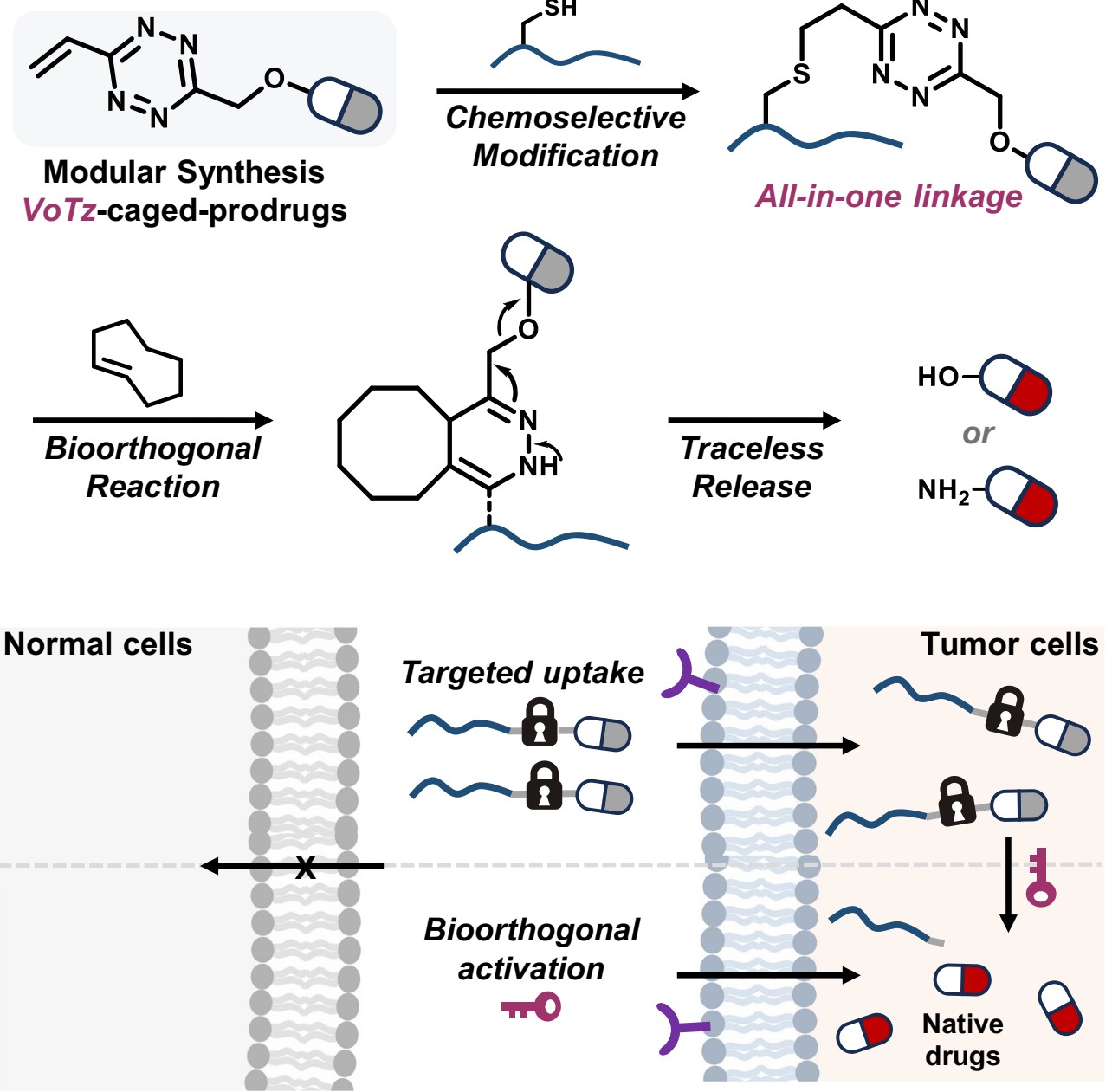

**Fig. 1 | The design of bioorthogonal activable PDC.** The design of all-in-one tetrazine reagent for chemoselective labeling and prodrug construction.

they would self-assemble. Interestingly, upon addition of peptide conjugates in aqueous solution, we observed an obvious Tyndall effect, indicating the existence of abundant nanostructures containing peptide conjugates (Fig. 3B and Supplementary Fig. 35). We investigated the morphologies and sizes of RGD-FL, RGD-SN-38 and RGD-Dox particles by dynamic light scattering (DLS) and transmission electron microscopy (TEM) analyses. The DLS results showed well-defined spherical morphologies with dark cores and average particle diameters of 108 nm (RGD-FL), 108 nm (RGD-SN-38), and 168 nm (RGD-Dox); these particle sizes were consistent with the TEM results (Fig. 3B and Supplementary Figs. 36 and 37). Additionally, these peptide conjugates were found to have relatively highly negative ζ-potentials (−23 to −21.5; Supplementary Fig. 38). Furthermore, by monitoring optical transmittance, the critical micellar concentrations (CMC) of RGD-SN-38 and RGD-Dox were determined to be 9.7 μM and 3.3 μM, respectively (Supplementary Fig. 39). These results suggested that the designed PDCs are inclined to self-assemble into single-component

nanomedicines[35] in a way that could potentially slow the biodegradation in the circulation and improve targeting efficiency.

We next proceeded to test whether the constructed peptide conjugates could remain stable in a biological milieu, so as to minimize the toxicity of the PDC to healthy tissues[7], and then release the payload on-demand via a bioorthogonal reaction at the target site. The thiol-exchange reaction is the most common side reaction in the Cys-based labeling approach[36], and reactions with biological thiols are the main reasons for tetrazine deactivation and degradation[17,37,38].

Therefore, we first investigated the stability of peptide conjugates in the presence of 10 mM of reduced glutathione (GSH) to mimic the high concentrations of reductive compounds in the cytoplasm. Compared with the rapidly degraded di- (pyridyl)tetrazine PyTz (degradation within 3 h), three representative peptide conjugates exhibited excellent stability against GSH-mediated degradation. After incubation at 37 °C for 12 h, we detected 96%,

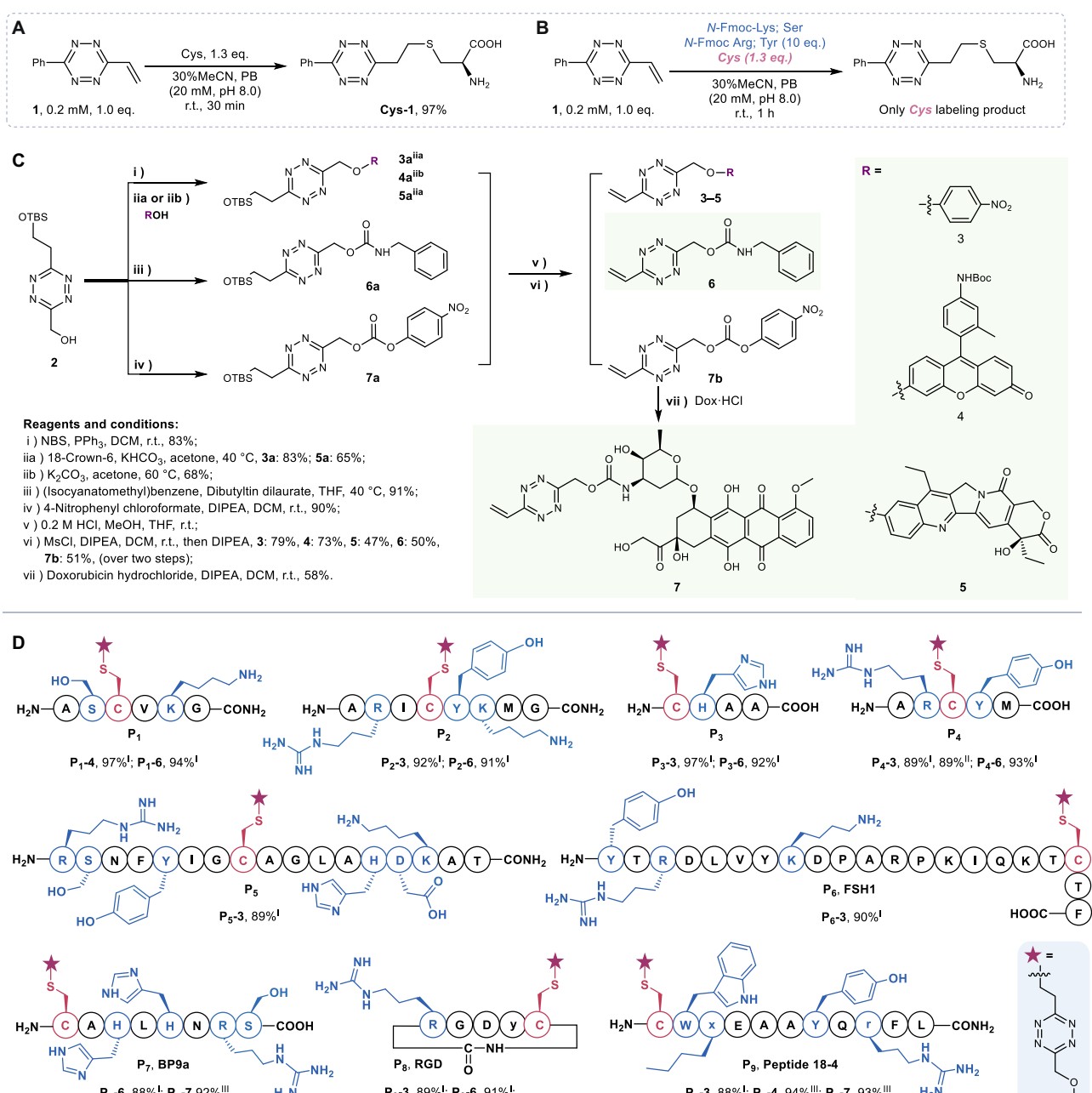

**Fig. 2 | Chemoselective modification of Cys and modular preparation of peptide conjugates using voTz linker reagents. A** The reaction of **1** with Cys provides the thioether product Cys-1, even when performed in the presence of a large excess of other nucleophilic amino acids (**B**). **C** Syntheses of voTz reagents. **D** Chemoselective preparation of peptide conjugates at the late stage. The yields were determined by relative integration of total ion counts based on LC–MS; I: the reactions were carried out under a 20 µM concentration of peptides in 20 mM PB (30% MeCN, pH 8.0); II: the reactions were carried out under a 5 µM concentration of peptides; III: the reactions were carried out under a 2.5 mM concentration of peptides, with 1.0 equiv. of DIPEA in MeCN: DMSO (3:2, v/v). Lower case letters denote d-amino acids. X is Nle.

92% and 85% of the RGD-Dox, RGD-SN-38 and RGD-FL, respectively, remained intact (Fig. 3C and Supplementary Fig. 40–44). In comparison, the pyrimidine tetrazine PmTz, which has been reported to be the most rapid trigger for the cleavage of allylic-substituted *trans*-cyclooctene (TCO)[39], was only 38% intact after a 12-h incubation with 10 mM GSH (Fig. 3C and Supplementary Fig. 45). Similarly, we found that 95% of RGD-Dox and 85% of RGD-SN-38 remained intact at 2 µM, below the critical micelle concentrations, after 12 h of incubation at 37 °C (Supplementary Fig. 46). This indicates that the observed stability is intrinsic to the PDCs, with a slight enhancement by micelle formation. In addition, the peptide conjugates were

found to be stable in serum, with 89% of RGD-SN-38 and 80% RGD-Dox found to be intact after 5 h (Supplementary Figs. 47 and 48).

We next selected three peptide conjugates, P₃−3 and P₃−6, and RGD-SN-38, to investigate the reaction kinetics of the tetrazine bioorthogonal reaction with TCO. By monitoring the decrease in tetrazine absorbance, we determined the second-order rate constants of P₃−3 and P₃−6 towards TCO to be 3277 ± 66 M⁻¹ s⁻¹ and 3184 ± 53 M⁻¹ s⁻¹, respectively, in PBS (pH 7.4) at 37 °C (Fig. 3D, Supplementary Figs. 49 and 50). RGD-SN-38 exhibited similar reactivity towards TCO, with a $k_2$ of 1705 ± 62 M⁻¹ s⁻¹ in PBS (pH 7.4, 10% DMF) at 37 °C (Supplementary Fig. 51). The high kinetics constants are consistent with

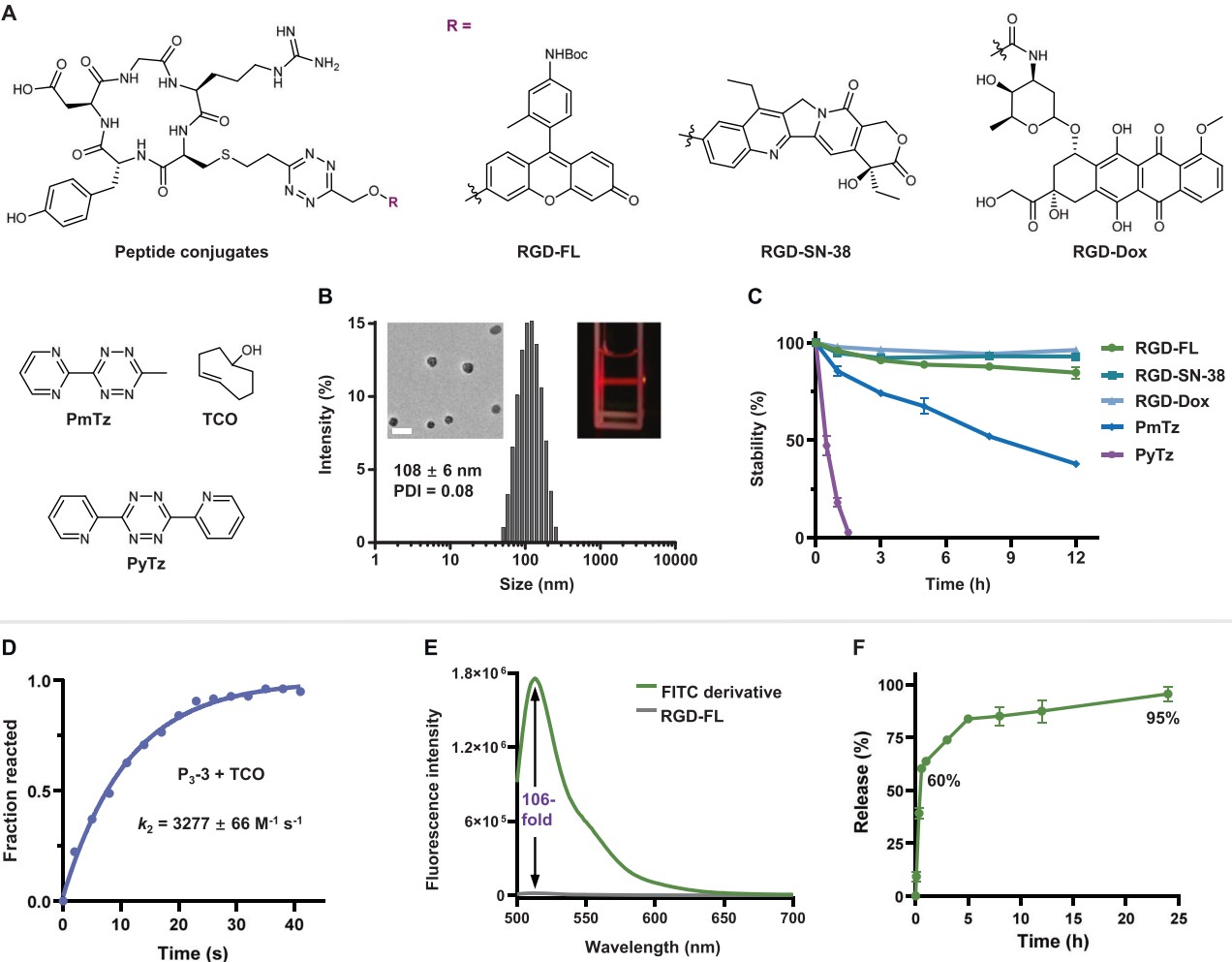

**Fig. 3 | Self-assembly and bioorthogonal click-to-release performance of peptide conjugates. A** Chemical structures of peptide conjugates and tetrazines (PyTz, PmTz) and TCO. **B** DLS histograms of RGD-FL in $H_2O$ (2% DMSO). Inserts are a TEM image of the corresponding self-assembly RGD-FL and a photograph of the corresponding self-assembly RGD-FL in $H_2O$ (2% DMSO) with a red laser passing through. Scale bar = 200 nm. Three times each experiment was repeated independently with similar results. **C** Stability of peptide conjugates, PmTz and PyTz under high GSH concentration (10% DMSO, PBS, pH 7.4, 37 °C, $n$ = 3 independent experiments) indicating the excellent stability of PDCs versus rapid degradation of PyTz and PmTz. **D** Kinetic studies of bioorthogonal reactions between $P_3$–3 (50 μM) with TCO (50 μM). **E** Fluorescence spectra of RGD-FL (2 μM, gray line) and unlinked fluorescein derivative (2 μM, green line) in PBS (1% DMSO), $\lambda_{ex}$ = 495 nm. **F** Rapid and complete release upon the bioorthogonal reaction between RGD-FL and TCO in PBS (1% DMSO, 37 °C, 50 μM RGD-FL + 500 μM TCO, $n$ = 3 independent experiments). All data are presented as mean±standard deviation (SD). Source data are provided as a Source Data file.

previous reports regarding dialkyltetrazines[40], and they signify sufficient bioorthogonal reactivity of tetrazine moiety for the release of drug at sub-micromolar concentrations. Considering that TCO analogs with higher reactivities can trigger the release from tetrazine with even faster kinetics[21], we envision that implementation of this strategy could bring the concentration down to the nanomolar range.

To assess the release reaction between peptide conjugates and the TCO trigger, we focused on the peptide-fluorophore conjugate RGD-FL, which also served as a fluorogenic probe[17,40–43]. Compared to the quenched conjugates, the unlinked fluorophore exhibited a 106-fold turn-on, enabling us to precisely monitor the click-to-release process both in solution and in live cells (Fig. 3E and Supplementary Fig. 52). Upon the addition of TCO, we observed that the instantaneous accomplishment of the bioorthogonal reaction, with cascade cleavage of ≃ 60% of the fluorophore occurring within 35 min, followed by a steady release segment achieving 95% release in 24 h (Fig. 3F and Supplementary Fig. 53). We hypothesize that the slower release rate observed in the later stage with was due to the tautomerization of dihydropyridazines, consistent with previous findings[21,22,44].

## Visualizing the activation of tumor-targeting peptide conjugates in live cells

Encouraged by the physiochemical tests, we next investigated the targeted release of payloads from peptide conjugates in live cells. Upon administering the bioorthogonal trigger TCO to $\alpha_v\beta_3$-positive SKOV3 ovarian cancer cells[45] pre-treated with RGD-FL, we detected a gradual increase in fluorescence signal in microscopic images, indicating an efficient click-to-release process (Fig. 4A, B and Supplementary Fig. 54). Examining the pixel intensity of fluorescence time course imaging led to the calculation of a release half-life of ≃ 16 min (Fig. 4C). Furthermore, confocal microscopic images clearly showed the fluorophore release taking place inside the cell (Fig. 4D and Supplementary Fig. 55). In contrast, control SKOV3 cells, which were not administered the TCO trigger or which had the $\alpha_v\beta_3$ target blocked by preincubation with a 20-fold excess of unmodified RGD peptide, exhibited negligible cellular fluorescence signals. Low fluorescence was also observed upon treatment of MCF-7 breast cancer cells, which exhibit low expression of integrin $\alpha_v\beta_3$[46] (Fig. 4E and Supplementary Fig. 56).

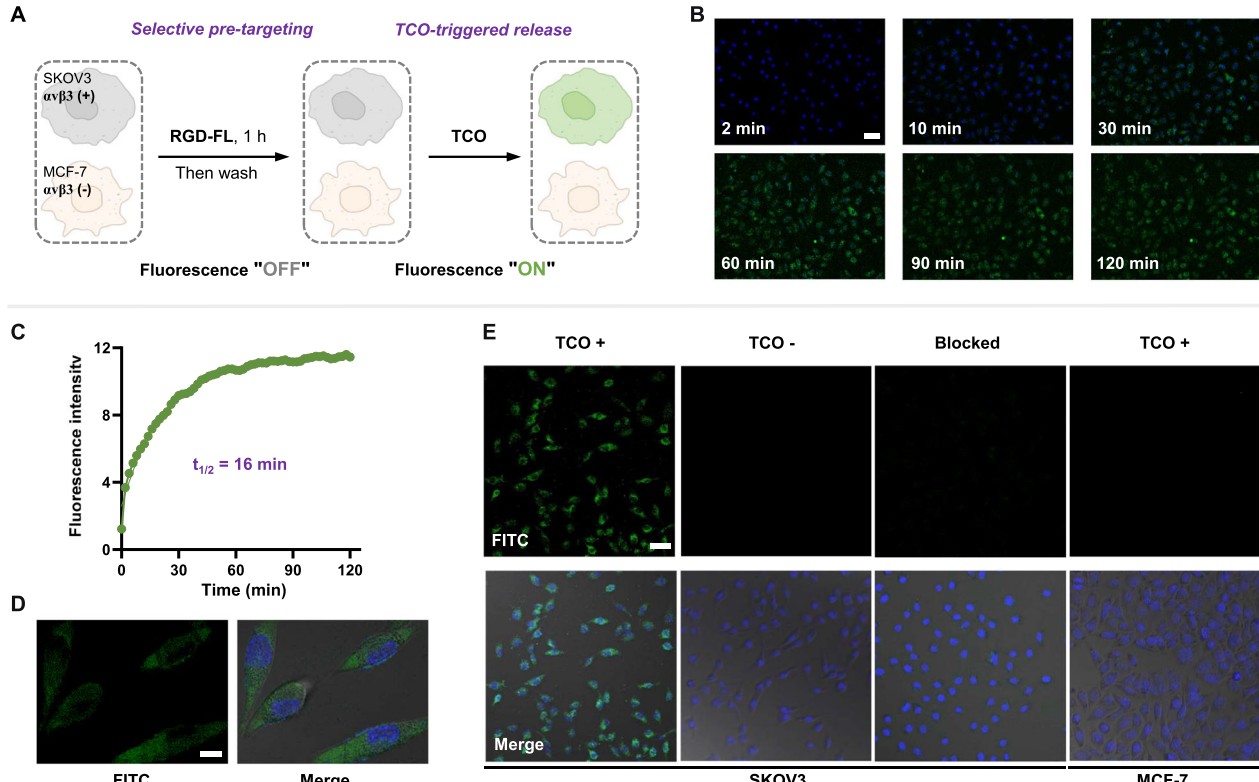

**Fig. 4 | Targeting and release of peptide conjugates in live cells. A** Protocol for visualizing the activation of tumor-targeting peptide conjugates using a TCO trigger. **B** Time-lapse imaging of RGD-FL release triggered by the bioorthogonal reaction in live SKOV3 cells. Scale bar: 50 μm. **C** Plot of pixel fluorescence intensity over time indicating the rapid RGD-FL bioorthogonal cleavage process in live cells. **D** Confocal image showing efficient uptake and intracellular release of RGD-FL.

Scale bar: 10 μm. **E** Live-cell imaging upon treatment using tumor-targeting peptide conjugates RGD-FL of $\alpha_v\beta_3$-positive SKOV3 cells and low-$\alpha_v\beta_3$ expressed MCF-7 cells (control). Blocking studies were performed by pre-treating SKOV3 cells with a 20-fold excess of unmodified RGD before incubation with RGD-FL and TCO. Scale bar: 50 μm. Three times each imaging experiment was repeated independently with similar results.

To further confirm the generality of targeted delivery by these peptide conjugates, we treated MDA-MB-231 breast cancer cells with 18-4-FL. Consistent with the bioorthogonal activation efficacy of RGD-FL in SKOV3 cell model, we detected rapid fluorophore release from conjugates inside live MDA-MB-231 cells upon treatment with TCO due to the bioorthogonal cleavage reaction (Supplementary Fig. 57). These observations suggest that the designed peptide conjugates can bind to a specific receptor on the targeting cell, successfully translocate the payload across the membrane, and activate bioorthogonally.

**Bioorthogonal activable PDC enables selective uptake with enhanced cytotoxicity**

To assess the synergy of combining chemotherapeutics with bioorthogonal activation of tumor-targeting PDCs, we conducted a study to investigate the antitumor effects of the activable PDC RGD-Dox in $\alpha_v\beta_3$-positive B16F10 melanoma cells, U87 brain glioma cells, and SKOV3 cells, as well as $\alpha_v\beta_3$-negative hepatic stellate (LX2) cells as a control. Unmodified Dox exhibited non-selective cytotoxicity against all four cell lines with IC$_{50}$ values ranging from 2.05 to 4.45 μM (Fig. 5A, blue line and Supplementary Table 2). The RGD-Dox prodrug exhibited significantly reduced cytotoxicity, especially for LX2 cells, with an IC$_{50}$ value of 56.73 μM, which was 17-fold higher than that of Dox (Fig. 5A, gray line and Supplementary Table 2). To determine the optimal time for TCO administration, we assessed the uptake of RGD-Dox by B16F10 cells at different time intervals using flow cytometry. The results revealed high cellular uptake after 2 h, leading us to select this time point for TCO administration to maximize intracellular release (Supplementary Fig. 58).

In contrast, when the RGD-Dox prodrug was activated by TCO, the cytotoxicity was increased 31- – 150-fold compared with the prodrug alone (Fig. 5B). The IC$_{50}$ values of activable-Dox dropped to 0.16 μM for B16F10, 0.95 μM for U87, and 1.08 μM for SKOV3 (Fig. 5A, red line and Supplementary Table 2). These values were 3.4- – 12.8-fold lower than direct administration of unmodified Dox, indicating the efficiency of this PDC design (Fig. 5B). Moreover, the activable-Dox showed significantly reduced toxicity against $\alpha_v\beta_3$-negative LX2 cells, with an IC$_{50}$ value of 16.42 μM, making the activable form 5-fold less toxic than Dox in LX2 cells, and 15.2- – 102.6-fold less toxic in $\alpha_v\beta_3$-positive cell lines (Fig. 5A, B). Furthermore, we also observed a similar increase of cytotoxicity for three other activable PDCs (BP9A-Dox, 18–4-Dox, and RGD-SN-38) towards their corresponding target tumor cells. These results demonstrate the generalizability of our strategy in improving the tumor specificity of chemotherapeutic agents (Supplementary Table 3 and Supplementary Fig. 59–63).

In addition, we verified that the mechanism of apoptosis induced by our prodrug release strategy was not different from that of Dox itself. B16F10 cells were treated with Dox and an equivalent amount of activable-Dox for 24 h, and total proteins were extracted and analyzed by western blotting. The treatment groups exhibited similar patterns of up-regulation of the apoptosis-related proteins p53, BAX, and caspase-3, suggesting that the fundamental reason for the cell-killing effect of our prodrug release strategy was the cytotoxicity caused by the released Dox (Supplementary Fig. 64).

In order to gain a better understanding of the enhanced tumor-specific cytotoxicity, we investigated the targeting and internalization efficacy of RGD-Dox inside live cells. Since Dox is fluorescent, with an

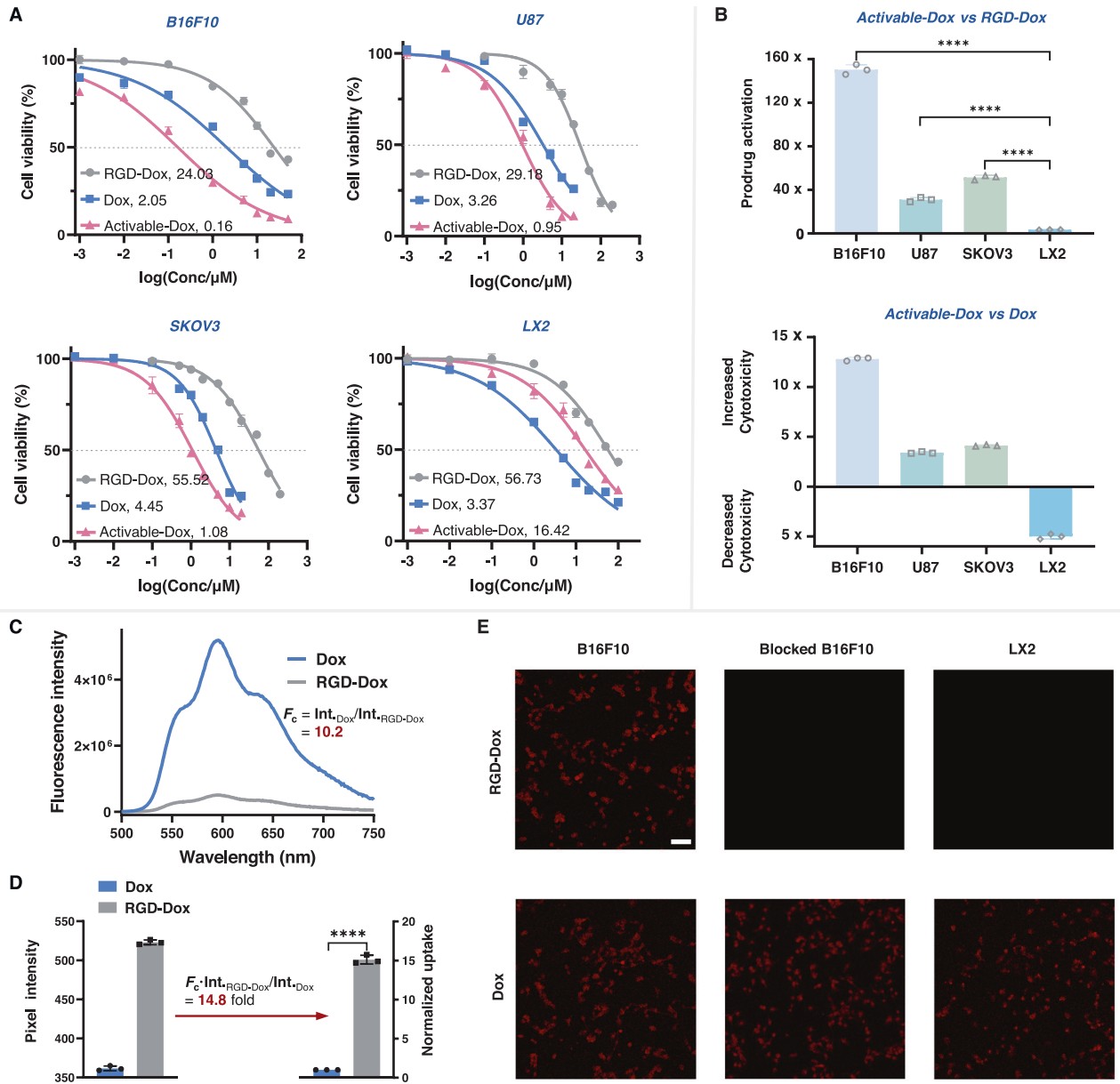

**Fig. 5 | The PDC strategy enhances the activity of Dox in cultured cells.**
**A** Cytotoxicity of Dox, RGD-Dox, and activable-Dox in B16F10, U87, SKOV3, and LX2 cells ($n = 3$ biologically independent samples). Gray and blue lines show cells incubated with RGD-Dox or Dox, respectively, for 48 h. The red lines indicate cells were pre-incubated with different concentrations of RGD-Dox for 2 h followed by incubation with TCO for 48 h in total. The insert table shows calculated $IC_{50}$ values (μM) for activable-Dox, Dox, and RGD-Dox against corresponding cells. **B** Prodrug activation (top), improved cytotoxicity and safety (bottom) by employing a bioorthogonal cleavage reaction of RGD-Dox ($n = 3$ biologically independent samples). Data were compared using one-way ANOVA followed by Tukey post-hoc test, ****$P < 0.0001$. **C** Fluorescence spectra of RGD-Dox (gray) and Dox (blue) at equal

concentrations (2 μM) in PBS (10% DMSO). **D** Fluorescence semi-quantification of the difference in uptake of RGD-Dox and Dox by B16F10 cells ($n = 3$ biologically independent samples). Differences between the two groups were assessed for significance using a two-tailed unpaired Student's $t$ test, ****$P < 0.0001$.
**E** Fluorescence imaging analysis of the uptake of RGD-Dox or Dox by B16F10 and LX2 cells. Cells were treated with 10 μM RGD-Dox or Dox for 2 h, then washed three times with PBS. Blocking studies were performed by pre-treatment of cells with a 50-fold excess of unmodified RGD before incubation with RGD-Dox. Scale bar: 100 μm. All data are presented as mean±SD. Source data are provided as a Source Data file.

emission maximum at approximately 595 nm[47], we visualized and quantified the cellular uptake level by monitoring pixel intensity. Considering the fluorescence quenching behavior of the tetrazine moiety, we first determined the ratio of fluorescence intensity of Dox to that of an equimolar amount of RGD-Dox in PBS (the fluorescence coefficient ($F_c$)) to be 10.2 (Fig. 5C and Supplementary Fig. 65).

Next, we cultured B16F10 cells with either Dox or RGD-Dox and measured the pixel intensity at 595 nm for each cell population in the absence of TCO. After correcting for the $F_c$, we found a 14.8-fold

increase in intensity in cells treated with RGD-Dox compared to those treated with Dox directly. This finding was consistent with the results of flow cytometry analysis (Fig. 5D and Supplementary Figs. 66 and 67), indicating a significantly higher intracellular concentration of Dox when using RGD-Dox. Furthermore, we observed negligible fluorescence in $\alpha_v\beta_3$-negative LX2 cells or in cells where endocytic pathways were blocked by pre-treatment with a 50-equiv. excess of unmodified RGD peptide. Conversely, cells treated with free Dox exhibited obvious uptake through nonspecific diffusion and internalization (Fig. 5E and

Supplementary Fig. 68). Taken together, these results indicate that the designed strategy enables biomarker-specific enhancement of the local concentration of the prodrug, and this phenomenon would be expected to have particular benefit in the context of hydrophobic cytotoxic drugs.

### Therapeutic efficacy of PDC in a melanoma tumor model

Next, we sought to evaluate the overall therapeutic potential of RGD-Dox using the subcutaneous B16F10 melanoma tumor model. Mice were randomly assigned to five groups ($n = 5$/group) once the tumor volume reached $\simeq$ 50–70 mm$^3$ on day 11 post-tumor inoculation. These groups underwent intravenous injections at 3-day intervals for a total of four administrations (Fig. 6A). The activable-Dox treatment group received a single dose of RGD-Dox (3.67 µmol·kg$^{-1}$), followed by the administration of biocompatible TCO[48] decorated with four PEG subunits (TCO-P$_4$, 36.7 µmol·kg$^{-1}$) after a 2 h interval. We chose this time interval because of the relatively high accumulation level of RGD-Dox at the tumor site at this time point. (Supplementary Fig. 69). For comparative analysis, the remaining four groups were subjected to tail intravenous

injections of RGD-Dox (3.67 µmol·kg$^{-1}$), TCO-P$_4$ (36.7 µmol·kg$^{-1}$), Dox (3.67 µmol·kg$^{-1}$), or PBS.

After four treatment courses, mice treated with RGD-Dox or TCO-P$_4$ alone showed rapid tumor growth that was similar to that of the PBS control group. The administration of Dox slightly slowed the tumor growth, but these mice experienced serious loss of body weight as compared with the PBS control group. In contrast, mice administered an equivalent dose of activable-Dox experienced a significant inhibition of tumor progression, with a tumor growth inhibition value (TGI) was as high as 87.6% (Supplementary Fig. 70), underscoring the success of this PDC strategy in vivo. In addition, mice treated with activable-Dox maintained a steady weight, similar to mice in the control group, further indicating the superiority of our strategy in terms of both efficacy and safety (Fig. 6B–D).

After the mice were sacrificed, the tumors and main organs were harvested. TUNEL staining and histological examination by hematoxylin-eosin staining showed that tumor tissues from the activable-Dox group exhibited much higher levels of necrosis and apoptosis (Fig. 6E, F). In the assessment of systemic toxicity, typical inflammatory injury[49] was observed in the cardiomyocytes and

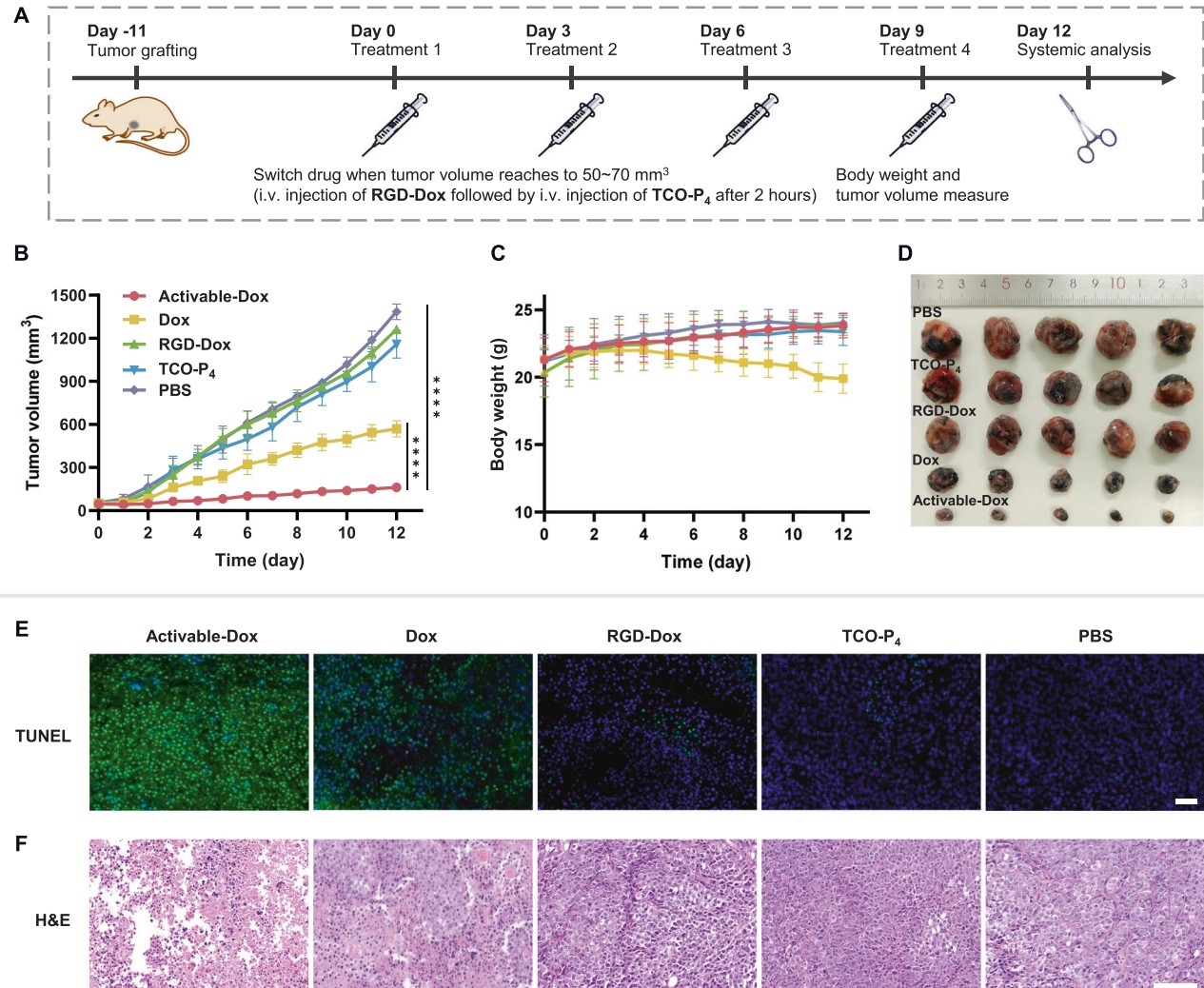

**Fig. 6 | In vivo anticancer activity in a melanoma tumor model. A** Flowchart showing the course of treatment. The mice received four treatment courses on days 0, 3, 6, and 9. Tumor size and body weight were measured every day. **B** Curve of the tumor volume change during treatment ($n = 5$ mice). Tumor volume was calculated as $V = 0.5 \times$ (minor axis) $\times$ (major axis)$^2$. Tumor volume data were compared using one-way ANOVA followed by Tukey post-hoc test, ****$P < 0.0001$.

**C** Changes in body weight ($n = 5$ mice). **D** Photographs of tumor samples dissected after treatment for 12 d. Representative histological examination results of the dissected tumors using (**E**) TUNEL staining and (**F**) H&E staining. Scale bar: 100 µm. All data are presented as mean ± SD. Source data are provided as a Source Data file.

pulmonary vessels of mice treated with Dox. In contrast, histological analyses of the heart, liver, spleen, lung, and kidney revealed no obvious differences between tissues of mice treated with activable-Dox and those of control mice treated with PBS, further indicating the good biocompatibility and safety of our prodrug strategy (Supplementary Fig. 71).

## Discussion

We have developed voTz as a versatile reagent for the modular preparation of bioorthogonal activable prodrugs. With this linker strategy, diverse pharmaceuticals or molecular probes can be selectively attached to Cys residues on targeting peptide under physiological conditions, yielding novel PDCs or peptide molecular probes with excellent yield. The resulting peptide conjugates self-assemble into nanostructure and remain stable within the biological environment. Integrating the selective and strong binding of peptides to targets and the exceptional bioorthogonal reactivity of the tetrazine linkage, this prodrug design strategy enables the selective and efficient release of the active drug into target cells. The fluorogenic feature of the bioorthogonal cleavage reaction also provides a convenient tool to visualize the release dynamics with spatio-temporal resolution and to quantify therapeutic uptake. In vitro studies showed that this approach significantly enhances therapeutic efficacy by substantially increasing the local drug concentration, therefore reducing the required dose. Furthermore, due to the avoidance of nonspecific diffusion and internalization, this PDC design strategy is expected to markedly reduce the systemic toxicity associated with conventional chemotherapeutic agents. A pilot animal study further confirmed tumor inhibition and safety and supported the promising therapeutic potency of these bioorthogonal activable PDCs.

Considering the wide functional group scope of bioorthogonal cleavage chemistry involving oxymethyl-tetrazine, we anticipate that the voTz designed in this study will be applicable to a wide array of cytotoxic molecules for the development of prodrugs, and it will be particularly useful for compounds with poor membrane penetration. Meanwhile, the vinyl group of voTz provides a reactive site for stoichiometric and chemoselective labeling not only for peptides but also for the smooth conjugation of tetrazine-caged molecules onto proteins, antibodies, and other thiol-containing macromolecules. We envision that this advancement will pave the way to the next generation of bioorthogonally activable biomedicines and biomaterials.

## Methods

### Chemoselective peptide labeling

HPLC conditions: Poroshell 120, EC-$C_{18}$ column: $3.0 \times 150$ mm, $2.7\,\mu$m; column temperature: 35 °C; gradient: 0 min, 5% B; 0–8 min, 5–100% B; flow rate: 0.5 mL/min.

Each voTz analog (1.1 equiv.) was added to a 0.25- mL vial containing solvent. The resulting solution was pipetted up and down five times, followed by the addition of a peptide solution ($P_1$–$P_9$) to a final concentration of either 20 μM or 2.5 mM in 200 μL. The solution was further pipetted up and down ten times and allowed to sit at room temperature. The chromatogram peak areas of all relevant Cys-containing species were integrated, and the yield was determined using the following equation: Yield (%) = $I_{product}$/($I_{starting} + I_{product} + I_{oxidation} + I_{side\ product}$) × 100, where $I_{starting}$, $I_{product}$, $I_{oxidation}$, and $I_{side\ product}$ respectively represent the average ion counts of remaining starting material, product, oxidized starting material and any side product, if present[50].

### Stability tests in the presence of a large excess of GSH

All tetrazine compounds were diluted to a concentration of 50 μM or 2 μM using PBS (10% DMSO, pH 7.4). A stock solution of GSH (400 mM in PBS) was then added to achieve a final concentration of 10 mM. The mixture was homogenized and incubated at 37 °C for

12 h. The stability of the compounds (including all peptide conjugates and PmTz) was assessed by measuring the decrease in peak area at 520 nm (with the initial peak area at 0 min defined as 100%). This measurement was monitored by HPLC–MS. Similarly, the stability of PyTz was determined by monitoring the decrease in peak intensity at 520 nm using UV–Vis. Statistical analysis was performed on data from three replicates to calculate the mean and standard deviation.

### Antibody

All the antibodies were obtained from Abcam Ltd (Cambridge, UK): Anti-p53 antibody, Rabbit polyclonal (ab131442, 1:500 dilution), Anti-Bax antibody, Rabbit monoclonal [E63] (ab32503, 1:4000 dilution), Anti-Caspase-3 antibody, Rabbit monoclonal [EPR18297] (ab184787, 1:2000 dilution), Anti-β-actin antibody, Rabbit polyclonal (ab8227, 1:4000 dilution), Anti-GAPDH antibody, Rabbit monoclonal [EPR16891] (ab181602, 1:10,000 dilution), Goat anti-Rabbit IgG HL (HRP, ab205718, 1:10,000 dilution).

### Cell lines

Human ovarian cancer SKOV3 cells (CL-0215), Human breast cancer MCF-7 cells (CL-0149), Human brain glioma U87 cells (CL-0238), Mouse melanoma B16F10 cells (CL-0319), Human hepatic stellate LX2 cells (CL-0560), Human hepatocellular carcinomas HepG2 cells (CL-0103) and Human breast cancer MDA-MB-231 cells (CL-0150) were kindly provided by Procell Life Science & Technology Co., Ltd. Authentication of all cells was conducted via short tandem repeat (STR) profiling in Procell Life Science & Technology Co., Ltd.

### Animal

Male BALB/c nude mice (five weeks old, 20 g) which were purchased from Beijing HFK Bioscience (Beijing, China) are housed at an ambient temperature of $23 \pm 2$ °C and relative humidities of $55\% \pm 2\%$ in a specific pathogen-free environment with a 12 h light/dark cycle. All animal experiments were approved (approval number: 20230625002) by the Committee for Animal Care and Use and the Ethics Committee of West China Hospital, Sichuan University.

### In vivo anticancer efficacy

Tumor-bearing mice were established by subcutaneously injecting a suspension of $5 \times 10^5$ B16F10 cells per mouse into the front flank of each nude mouse. When tumor volume reached 50–70 $mm^3$, the animals were randomly divided into five groups ($n = 5$). The mice were administered different compounds suspended in PBS via the tail vein every 3 days for a total of four times. The treatment group received tail intravenous injection of RGD-Dox (3.67 μmol·$kg^{-1}$, 1.0 eq.) and TCO-$P_4$ (36.7 μmol·$kg^{-1}$, 10 eq.) after 2 h intervals to activate Dox (abbreviated as activable-Dox).

For comparison, the remaining four control groups received tail intravenous injections of RGD-Dox (3.67 μmol·$kg^{-1}$), TCO-$P_4$ (36.7 μmol·$kg^{-1}$), Dox (3.67 μmol·$kg^{-1}$, 1.0 eq.), or PBS. The mice underwent four treatment courses on day 0, day 3, day 6, and day 9. Tumor size and body weight were measured every day. Tumor volume was calculated via the following formula: V = 1/2 × (minor axis) × (major axis)[2]. The volume of all tumors did not exceed the maximum permitted by the ethics committee (2000 $mm^3$).

After 12 days, the mice were sacrificed and dissected to obtain tumor tissues and main organs, including heart, liver, spleen, lung, and kidney. Subsequently, all tissues were fixed in a 4% paraformaldehyde solution, routinely paraffin-embedded and sectioned, and then stained with hematoxylin and eosin, as well as using TdT-mediated dUTP nick-end labeling (TUNEL) for subsequent pathological analysis. TUNEL-positive cells appeared fluorescent green, representing the apoptosis of tumor cells (nuclei were fluorescent blue, stained with DAPI).

## Reporting summary

Further information on research design is available in the Nature Portfolio Reporting Summary linked to this article.

## Data availability

The authors declare that the data supporting the findings of this study are available with the paper and its Supplementary information files. Source data are provided in this paper.

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

## Acknowledgements

This work was supported by the National Natural Science Foundation of China (22177081 to H.W., 52273136 to H.S.), the National Key R&D Program of China (2022YFC2009902 to H.W.), the Sichuan Science and Technology Program (2023YFH0071 to H.S.) and the 1•3•5 Project for Disciplines of Excellence at West China Hospital, Sichuan University (ZZYC23003 to H.W.). Med-X Innovation Program of Med-X Center for Materials, Sichuan University (MCM202304 to H.W. and H.S.). This work was also financially supported by the State Key Laboratory of Polymer Materials Engineering (Grant No.: sklpme2022-3-14 to H.S.). We thank Feijing Su and Qifeng Liu at the Core Facilities of West China Hospital and the Analytical & Testing Center of Sichuan University for their help with NMR measurements.

## Author contributions

H.W. and X.H. initiated and designed the project. X.H., J.L., X.L., W.M., X.D., M.Q. and H.S. carried out all the experiments and analyzed the experimental data. H.W. and X.H. prepared the manuscript. All authors discussed and commented on the manuscript.

## Competing interests

The authors declare no competing interests.
