## [Peer Review File · Nature Communications]

An All-in-One Tetrazine Reagent for Cysteine-Selective Labeling and Bioorthogonal Activable Prodrug ConstructionREVIEWER COMMENTS

Reviewer #1 (Remarks to the Author):

The authors report a novel approach to link chemotherapeutic drugs to cancer cell targeting ligands, such as peptides. The linking (conjugation) is done with a vinyltetrazine (Tz) linker, which allows chemoselective modification of a thiol group in the presence of other unprotected functional groups, like amine, alcohol, etc., in the targeting peptide. In addition, the Tz linker is cleaved with an exogenous agent (TCO), while currently most peptide-drug conjugates (PDCs) utilize endogenous stimuli to release the drug from the targeting peptide.

The chemoselective modification of a free thiol (from Cys) in native peptides under mild conditions using the Tz linker is novel. This would be widely applicable for the synthesis of PDCs, and likely other conjugates, that serve as carrier-linked prodrugs for targeted drug delivery to cancer cells via overexpressed cell-surface receptors. The manuscript is well written, and I recommend this study for publication after addressing the following comments.

Comments

Page 2, 2nd paragraph “overexpressed in the tumor microenvironment⁵”: In addition, a recent review on linkers that respond to specific endogenous stimuli could be cited here - Alas M. et al. *J. Med. Chem.* 2021, 216.

Page 3, 2nd paragraph: Please elaborate on the synthesis of linking oxymethyl moiety with the drug which is required for the Tz approach.

Page 6, 1st paragraph: Peptide 18-4 targets keratin 1 in breast cancer cells (Soudy R. et al. *Mol. Pharm.* 2017, 593). This should be added.

Page 11, Figure 4A: Explain here why the cytotoxicity of Dox and Activable-Dox differs in $\alpha\beta$ 3-positive cells. Why is the IC₅₀ several-fold lower for Activable-Dox compared to free

Dox?

Page 11, Figure 4A: Why cells were incubated with TCO 2 hours after treating the cells with RGD-Dox. The reason for picking 2 hours should be explained here.

Page 13, 1st paragraph: Why was TCO administered 2 hours after RGD-Dox in mice? Or how was the time determined for the administration of TCO in mice? If TCO is administered a bit early or several hours after RGD-Dox, will the approach still work? Could this be a limitation? Please elaborate on this.

Finally, there are some grammatical errors and typos in the text. For instance, page 4, 2nd paragraph reads as “and elimination afforded produced the desired voTz-caged ethers”, page 6, line 2, “tumor-tageting peptides”. There should be careful reading and editing of the manuscript before publication.

Reviewer #2 (Remarks to the Author):

This manuscript by Wu et al. presents an integrated strategy for the modular development of activable prodrugs. The in vivo therapeutic performance of this strategy is explored using bioorthogonal cleavage chemistry to release drugs from tetrazine derivatives. The newly developed 3-vinyl-6-oxymethyl-tetrazine serves as both a chemoselective reagent for labeling Cys residue and a bioorthogonal reactivable functional group for prodrug design. The prodrugs exhibit good stability, bioorthogonal reactivity, and intriguing self-assembly behaviors. A systematic in vitro and in vivo study demonstrates improved therapeutic efficacy and reduced systemic toxicity compared with traditional chemotherapeutic agents. Overall, this article is of high quality and addresses a topic highly relevant to the community, encompassing methodology design, prodrug investigations, and in vivo validation. The writing style is appropriate and concise, the interpretation aligns with the results obtained, and the methods presented are thorough and interdisciplinary. I recommend publication in Nature Communications after the following revisions:

1. Please provide information on the reaction kinetics of vinyl-tetrazine reagents in the

labeling step. While the authors demonstrated the labeling reaction at 20 μM , it is essential to include data on reaction yield at lower concentrations.

2. In the visualization of prodrug activation, please provide relevant details, including the analysis wavelength, for measuring fluorescent density in live cells.

3. For the prodrug selective uptake study, it would be better to quantify prodrug uptake through flow cytometry in the prodrug selective uptake study.

4. In Figure 5, elaborate on the observation that the body weights of the activable-DOX group show no significant difference compared to PBS, RGD-DOX, and TCO-P4 groups, despite much smaller tumor volumes. Please provide tumor weights and calculate the tumor inhibitory rate to address this discrepancy.

5. The reference list can be improved. The #9 ref is not formatted in style. Some on-target bioorthogonal prodrug activation work should be cited, e.g. *Nat. Commun.*, 2018, 9, 5032; *Bioconjugate Chem.* 2023, doi: 10.1021/acs.bioconjchem.3c00404.

6. Several ^1H NMR spectra exhibit peaks around 1.6 ppm which were not attributed (e.g., compound 1). It is suggested that the author provides a proper description of the substance associated with these peaks.

Reviewer #3 (Remarks to the Author):

This contribution by He et al describes an elegant prodrug design strategy that takes advantage of a drug release mechanism triggered by bioorthogonal conjugation. This release mechanism was first reported by Robillard in 2013 (ref 18) and then optimized by Robillard in 2020 (ref 20) and the authors of this contribution in 2021 (ref 21). The current submission applies this drug release mechanism to designing peptide -drug conjugates (PDCs), which have shown promise as privileged therapeutics. Specifically, the authors report that vinyl substituted tetrazines show facile conjugation to cysteine thiols, while maintaining their bioorthogonal conjugation with TCOs for drug release (Figure 2). The drug release mechanism has been realized on cultured cells (Figure 3-4) as well as in a mouse model of cancer (Figure 5). Particularly noteworthy is that the prodrugs (peptide-drug conjugates) affords cancer cell target capabilities, and upon activation (drug release) via TCO conjugation, kills cancer cells more efficiently than the corresponding drug-only controls. This is a creative study with carefully designed experiments to support the authors'

conclusions. The significance of this work lies in the peptide-drug conjugation methodology (via vinyl tetrazine) as well as the *in vivo* demonstration of the power of the click-and-release strategy for designing peptide-drug conjugates as prodrugs. This work would be suitable for publication in Nature Comm after the following issues are addressed:

1. The observation of nanostructure formation (Figure 2) is interesting. However, the origin of the observed stability is not fully clear. Has the author examined the stability of the peptide-drug conjugate at below its critical micelle concentrations? This experiment will teach if the observed stability is intrinsic of the peptide-drug conjugates or due to micelle formation.
2. The drug release appears to exhibit a two-phase mechanism (Figure 2F). A fast phase of 35 min and then a slow phase of 24 hrs. How should one understand and rationalize this two-phase behavior?

Response to Reviewer 1:

The authors report a novel approach to link chemotherapeutic drugs to cancer cell targeting ligands, such as peptides. The linking (conjugation) is done with a vinyltetrazine (Tz) linker, which allows chemoselective modification of a thiol group in the presence of other unprotected functional groups, like amine, alcohol, etc., in the targeting peptide. In addition, the Tz linker is cleaved with an exogenous agent (TCO), while currently most peptide-drug conjugates (PDCs) utilize endogenous stimuli to release the drug from the targeting peptide.

The chemoselective modification of a free thiol (from Cys) in native peptides under mild conditions using the Tz linker is novel. This would be widely applicable for the synthesis of PDCs, and likely other conjugates, that serve as carrier-linked prodrugs for targeted drug delivery to cancer cells via overexpressed cell-surface receptors. The manuscript is well written, and I recommend this study for publication after addressing the following comments.

Comments 1: Page 2, 2nd paragraph “overexpressed in the tumor microenvironment⁵”: In addition, a recent review on linkers that respond to specific endogenous stimuli could be cited here - Alas M. et al. *J. Med. Chem.* 2021, 216.

Response 1: We appreciate your comment. We have included reference 12 in the second paragraph on page 2 as follows:

“To fulfill this requirement, researchers have developed various cleavable linkers that respond to specific stimuli such as pH, reducing potential or enzymes that are overexpressed in the tumor microenvironment^{5, 12.}”

12. Alas, M., Saghaidehkordi, A., Kaur, K. Peptide-Drug Conjugates with Different Linkers for Cancer Therapy. *J. Med. Chem.* **64**, 216–232 (2021).

Comments 2: Page 3, 2nd paragraph: Please elaborate on the synthesis of linking oxymethyl moiety with the drug which is required for the Tz approach.

Response 2: We appreciate your insightful suggestion. To enhance the clarity of our presentation, we have provided more comprehensive details on these aspects within the manuscript. The revised text is thoughtfully presented below for your convenience:

To address these questions, we designed a key precursor, hydroxymethyl-tetrazine (**2**), which was prepared in three steps from commercially available materials on a multigram scale with an overall yield of 20% (see details in the Supplementary Information). Starting with compound **2**, we conducted NBS bromination and alkylation to obtain three phenol ether intermediates (**3a–5a**) in a straightforward manner. After removing the TBS group, a cascade mesylation and elimination produced the desired **voTz**-caged ethers (**3–5**) in moderate to good yields (Figure 1C, 47–79%). The versatile precursor **2** could be converted to carbamate **6a** in a 91% yield by treating it with isocyanate. Additionally, it could be transformed into an active carbonate (**7a**) in 90% yield. Following the same deprotection and cascade mesylation and elimination procedure, we successfully obtained the **voTz**-caged carbamate **6** and **voTz**-caged carbonate ester **7b** in moderate yields (Figure 1C, 50–51%). With an aminolysis step involving the reaction of **7b** with doxorubicin, we formed the desired carbamate **7** in good yield. Consequently, we

obtained five **voTz** derivatives, including two model compounds (**3** and **6**), caged fluorophore **4**, and two prodrugs containing a camptothecin analogue (SN-38, **5**) or doxorubicin (Dox, **7**).

Comments 3: Page 6, 1st paragraph: Peptide 18-4 targets keratin 1 in breast cancer cells (Soudy R. et al. *Mol. Pharm.* 2017, 593). This should be added.

Response 3: We appreciate your comment. We have incorporated reference 35 into the first paragraph of page 6, as follows:

“**P₉** (peptide 18-4), which targets keratin 1 in breast cancer cells^{34, 35}”

35. Soudy, R., Etayash, H., Bahadorani, K., Lavasanifar, A., Kaur, K. Breast Cancer Targeting Peptide Binds Keratin 1: A New Molecular Marker for Targeted Drug Delivery to Breast Cancer. *Mol. Pharmaceutics* **14**, 593–604 (2017).

Comments 4: Page 11, Figure 4A: Explain here why the cytotoxicity of Dox and Activable-Dox differs in $\alpha_v\beta_3$ -positive cells. Why is the IC50 several-fold lower for Activable-Dox compared to free Dox?

Response 4: We appreciate your insightful suggestion. Firstly, for different $\alpha_v\beta_3$ -positive cells, variations in the expression levels and membrane permeability contribute to distinct IC50 values for Dox and Activable-Dox.

Concerning the lower IC50 of Activable-Dox compared to free Dox in the same cell line, the explanation lies in the prodrug RGD-Dox, which incorporates a tumor-targeting peptide to enhance cellular uptake. To bolster the credibility of our cell uptake experiments, we conducted flow cytometry analysis. The results revealed that for $\alpha_v\beta_3$ -positive cells, the uptake of RGD-Dox was 20.8 times higher than that of Dox, aligning with the findings from fluorescence semi-quantitative uptake experiments (Figure 4, C–E).

Further elaborating on this discovery in our manuscript, we stated: “This finding was consistent with the results of flow cytometry analysis (Figures 4D and S66, S67), indicating a significantly higher intracellular concentration of Dox when using **RGD-Dox**.”

Detail information regarding these experiments is included in the Supplementary Information (Page S53), as presented below:

16.3 Comparison of cell uptake between RGD-Dox and Dox using flow cytometry

The cellular uptake of **RGD-Dox** and Dox was analyzed using flow cytometry. B16F10 cells, characterized by $\alpha_v\beta_3$ positivity, were seeded onto 6-well cell culture clusters and incubated for 48 hours. Following the incubation period, the cells were exposed to either **RGD-Dox** (10 μ M), Dox (10 μ M), or PBS at 37 °C for 2 hours. Subsequently, the cells were trypsinized and underwent three PBS rinses. Flow cytometry analysis (Fortessa, BD, USA) was utilized to evaluate the intrinsic fluorescence intensity of Dox with an excitation/emission wavelength of 488/610 nm.

Supplementary Fig. 67. Analysis of cell uptake through flow cytometry. (a–c) B16F10 cells were incubated with various treatments (10 μ M) at 37 $^{\circ}$ C for 2 h. (d) Quantification of fluorescence reveals the disparity in uptake between **RGD-Dox** and Dox by B16F10 cells. Error bars represent the standard deviation (n = 3).

Comments 5: Page 11, Figure 4A: Why cells were incubated with TCO 2 hours after treating the cells with RGD-Dox. The reason for picking 2 hours should be explained here.

Response 5: We appreciate your insightful comments. To provide further clarify regarding the rationale behind administering TCO 2 hours after treating the cells with **RGD-Dox**, we conducted flow cytometry analysis to assess the uptake of **RGD-Dox** by B16F10 cells at various time points (0.5 h, 1h, 2h, 4h, and 6h). The results demonstrated a time-dependent increase in **RGD-Dox** uptake, with a rapid increase observed within the first 2 hours, followed by a plateau and saturation thereafter. Considering the need to minimize unnecessary degradation of **RGD-Dox**, we identified the 2-hour time point as optimal for TCO administration.

In our manuscript, we expanded upon this observation, stating, "To determine the optimal time for TCO administration, we assessed the uptake of **RGD-Dox** by B16F10 cells at different time intervals using flow cytometry. The results revealed high cellular uptake after 2 hours, leading us to select this time point for TCO administration to maximize intracellular release (Figure S58)."

Furthermore, we have included detailed information about these experiments in the Supplementary Information

(Page S49), as outlined below:

15.1 Analysis of cellular uptake of RGD-Dox over various time intervals using flow cytometry

B16F10 cells were seeded onto 6-well cell culture clusters and incubated for 48 hours. After the incubation period, the cells were exposed to RGD-Dox (10 μ M) at 37 $^{\circ}$ C for different durations. Subsequently, the cells were trypsinized and underwent three rinses with PBS. Flow cytometry analysis (Fortessa, BD, USA) was utilized to evaluate the intrinsic fluorescence intensity of Dox with an excitation/emission wavelength of 488/610 nm.

Supplementary Fig. 58. Flow cytometry assessment of cellular uptake of RGD-Dox by B16F10 cells at different time points. Error bars represent the standard deviation (n = 3).

Comments 6: Page 13, 1st paragraph: Why was TCO administered 2 hours after RGD-Dox in mice? Or how was the time determined for the administration of TCO in mice? If TCO is administered a bit early or several hours after RGD-Dox, will the approach still work? Could this be a limitation? Please elaborate on this.

Response 6: We appreciate your insightful suggestion. In the initial experiments, we administered TCO 2 hours after RGD-Dox in mice based on the results from in vitro cell experiments. To determine the optimal administration time for TCO injection, we assessed the accumulation of RGD-Dox at the tumor tissue of B16F10 tumor-bearing nude mice at different time points (0-6 hours) after the tail vein injection. The results showed a continuous increase in the accumulation of RGD-Dox at the tumor site within the first 0-2 hours, followed by a gradual decrease. While we believe the approach would still work if TCO is administered a bit early or several hours after RGD-Dox, considering the relatively higher accumulation of RGD-Dox within approximately 2 hours in the tumor tissue, we believe that administering TCO 2 hours after RGD-Dox in mice is an appropriate time window.

We have included the details of the aforementioned experiments in the Supplementary Information (Pages S54–S55), as outlined below:

17.1 The accumulation of **RGD-Dox** at various time points in tumor tissues

When the tumor volume reached approximately 400 mm³, five mice in each group received an i.v. injection of 3.67 μmol·kg⁻¹ **RGD-Dox**. After various time intervals, the mice were sacrificed, and tumor tissues were dissected. About 0.5 to 1 g of tissue was ground into a homogenate, and proteins were precipitated by the adding two volumes of acetonitrile (1 mL). After centrifugation, the supernatant was removed, concentrated to 100 μL, and 20 μL was used for HPLC–MS analysis. The **RGD-Dox** concentration in the sample was determined from a standard curve of the peak area of the characteristic absorption of **RGD-Dox** at 520 nm.

Supplementary Fig. 69. (a) The standard absorption curve of **RGD-Dox** at various concentrations. (20 μL of sample was injected.) (b) The average concentration of **RGD-Dox** (3.67 μmol·kg⁻¹) in tumor after i.v. injection at various time intervals. Error bars represent the standard deviation (n = 5)

Meanwhile, in our manuscript, we have further elaborated on this finding: “The **activable-Dox** treatment group received a single dose of **RGD-Dox** (3.67 μmol·kg⁻¹), followed by the administration of biocompatible TCO⁴⁹ decorated with four PEG subunits (**TCO-P₄**, 36.7 μmol·kg⁻¹) after a 2-hour interval. We chose this time interval because of the relatively high accumulation level of **RGD-Dox** at the tumor site at this time point. (Figure S69).”

Comments 7: Finally, there are some grammatical errors and typos in the text. For instance, page 4, 2nd paragraph reads as “and elimination afforded produced the desired voTz-caged ethers”, page 6, line 2, “tumor-tageting peptides”. There should be careful reading and editing of the manuscript before publication.

Response 7: Thank you for your valuable feedback. We have carefully reviewed the manuscript and addressed the highlighted grammatical errors in the manuscript:

“and elimination produced the desired **voTz**-caged ethers”; “tumor-targeting peptides”

Your assistance in refining the manuscript is greatly appreciated.

Response to Reviewer 2:

This manuscript by Wu et al. presents an integrated strategy for the modular development of activable prodrugs. The in vivo therapeutic performance of this strategy is explored using bioorthogonal cleavage chemistry to release drugs from tetrazine derivatives. The newly developed 3-vinyl-6-oxymethyl-tetrazine serves as both a chemoselective reagent for labeling Cys residue and a bioorthogonal reactive functional group for prodrug design. The prodrugs exhibit good stability, bioorthogonal reactivity, and intriguing self-assembly behaviors. A systematic in vitro and in vivo study demonstrates improved therapeutic efficacy and reduced systemic toxicity compared with traditional chemotherapeutic agents.

Overall, this article is of high quality and addresses a topic highly relevant to the community, encompassing methodology design, prodrug investigations, and in vivo validation. The writing style is appropriate and concise, the interpretation aligns with the results obtained, and the methods presented are thorough and interdisciplinary. I recommend publication in Nature Communications after the following revisions:

Comments 1: Please provide information on the reaction kinetics of vinyl-tetrazine reagents in the labeling step. While the authors demonstrated the labeling reaction at 20 μM , it is essential to include data on reaction yield at lower concentrations.

Response 1: Thank you for your valuable comment. Firstly, we investigated the reaction kinetics of the labeling step between vinyl tetrazine (voTz) **1** and Cys, obtaining a second-order rate constant of $13 \pm 0.8 \text{ M}^{-1} \text{ s}^{-1}$. In our manuscript, we provide detailed information on this finding: "Additionally, a second-order rate constant of $13 \pm 0.8 \text{ M}^{-1} \text{ s}^{-1}$ was determined for the labeling reaction between **1** and Cys (Figure S12)."

We have included the details of these experiments in the Supplementary Information (Page S16), as outlined below:

5. Determine labeling reaction kinetics between voTz **1** and Cys

The second-order rate constant for the labeling reactions between tetrazine **1** and Cys was determined at 25 °C in PB (30% MeCN, 20 mM, pH 8.0) under second-order conditions using a 6000+ (Quawell) UV-Vis spectrophotometer. The concentrations of cysteine and tetrazine were identical. The specific difference in absorption wavelength (300 nm) between **1** and Cys-1 was measured over time. The second-order rate constant k_2 was calculated from the slope of a plot of $(1/c - 1/c_0)$ versus time.

Supplementary Fig. 12. (a) Absorption spectra of **1** (200 μM) and Cys-1 (200 μM) in PB (20 mM, pH = 8.0, 30% MeCN). The specific difference in absorption wavelength (300 nm) between **1** and Cys-1 was used to measure reaction kinetics. (b) The

absorption change over time of the reaction between **1** (50 μM) and Cys (50 μM) at 300 nm. (c) Calculation of the second-order rate constant of **1** (50 μM) and Cys (50 μM) by measuring the absorbance at 300 nm.

Additionally, we explored the labeling yield at lower concentrations (5 μM) between **voTz 3** and peptide **P₄**, achieving the same labeling yield as the higher concentration (20 μM). We made modifications to Figure 1 in the main text and provided more comprehensive details on these aspects within the manuscript. The revised text and Figure 1 are presented below for your convenience.

“The conjugation can also be smoothly performed at lower concentrations under the same conditions with yield of 89%, or at high concentrations in organic solvent for scale-up with same high yields (91–94%).”

Figure 1. Chemoselective modification of Cys and modular preparation of peptide conjugates using **voTz** linker reagents. (A) The reaction of **1** with Cys provides the thioether product **Cys-1**, even when performed in the presence of a large excess of other nucleophilic amino acids (B). (C) Syntheses of **voTz** reagents. (D) Chemoselective preparation of peptide conjugates at the late stage. The yields were determined by relative integration of total ion counts based on LC–MS; I: the reactions were carried out under a 20 μ M concentration of peptides in 20 mM PB (30% MeCN, pH 8.0); II: the reactions were carried out under a 5 μ M concentration of peptides; III: the reactions were carried out under a 2.5 mM concentration of peptides, with 1.0 equiv. of DIPEA in MeCN:DMSO (3:2, v/v). Lower case letters denote D-amino acids. X is Nle.

We have also included the details of the aforementioned experiments in the Supplementary Information (Page S25), as shown below:

Supplementary Fig. 21. (a) The labeling reaction involving **P₄** and **3** at 5 μ M. (b) HPLC–MS chromatogram of the reaction. (c) The associated mass spectrum of the product **P₄-3**.

Comments 2: In the visualization of prodrug activation, please provide relevant details, including the analysis wavelength, for measuring fluorescent density in live cells.

Response 2: We thank the reviewer's comment. We have included details about the analysis wavelength in Supplementary Information (page 46), as shown below:

14.1 Time lapse imaging of bioorthogonal reaction triggered **RGD-FL** releasing in live SKOV3 cells

SKOV3 cells were seeded onto 12-well cell culture clusters and cultured for 48 h. Subsequently, the cells were

stained with Hoechst 33342 (250 nM) for 5 min to label the nuclei, washed briefly, then pre-treated with **RGD-FL** (5 μ M) for 1 h at 37 °C. After washing with PBS, TCO (50 μ M) was added to the cells, which were then imaged using a confocal laser scanning microscope. Images were analyzed with ZEN blue software to measure the intensity (λ_{ex} = 488 nm, λ_{em} = 500–600 nm).

Comments 3: For the prodrug selective uptake study, it would be better to quantify prodrug uptake through flow cytometry in the prodrug selective uptake study.

Response 3: We appreciate your insightful comment. To strengthen the credibility of our prodrug selective uptake study, we employed flow cytometry to quantify cellular uptake. The results demonstrated a remarkable 20.8-fold increase in the uptake of RGD-Dox compared to Dox, specifically in $\alpha_v\beta_3$ -positive cells. This observation aligns with the results of our fluorescence semi-quantitative uptake experiment.

To provide additional support for this finding, we have expounded upon the results in our manuscript, stating, “This finding was consistent with the results of flow cytometry analysis (Figures 4D and S66, S67), indicating a significantly higher intracellular concentration of Dox when using **RGD-Dox**.”

We have also included the details of the aforementioned experiments in the Supplementary Information (Page S53), as shown below:

*16.3 Comparison of cell uptake between **RGD-Dox** and Dox using flow cytometry*

The cellular uptake of **RGD-Dox** and Dox was analyzed using flow cytometry. B16F10 cells, characterized by $\alpha_v\beta_3$ positivity, were seeded onto 6-well cell culture clusters and incubated for 48 hours. Following the incubation period, the cells were exposed to either **RGD-Dox** (10 μ M), Dox (10 μ M), or PBS at 37 °C for 2 hours. Subsequently, the cells were trypsinized and underwent three PBS rinses. Flow cytometry analysis (Fortessa, BD, USA) was utilized to evaluate the intrinsic fluorescence intensity of Dox with an excitation/emission wavelength of 488/610 nm.

Supplementary Fig. 67. Analysis of cell uptake through flow cytometry. (a–c) B16F10 cells were incubated with various treatments (10 μ M) at 37 $^{\circ}$ C for 2 h. (d) Quantification of fluorescence reveals the disparity in uptake between **RGD-Dox** and **Dox** by B16F10 cells. Error bars represent the standard deviation (n = 3).

Comments 4: In Figure 5, elaborate on the observation that the body weights of the activable-DOX group show no significant difference compared to PBS, RGD-DOX, and TCO-P4 groups, despite much smaller tumor volumes. Please provide tumor weights and calculate the tumor inhibitory rate to address this discrepancy.

Response 4: Thank you for your comment. In case of the activable-Dox group, tumor inhibition was effectively achieved, allowing nude mice to maintain a healthy weight. In contrast, the control group exhibited poor tumor inhibition, leading to competition for nutrients between the tumor and normal tissues. As a result, there was slow overall weight growth but obvious tumor expansion, resulting in similar overall weights between these groups.

To address this discrepancy, we have calculated the tumor inhibition rate. After the completion of the treatment cycle, the activable-Dox group demonstrated a remarkable tumor inhibition rate of 87.6%, while the Dox group showed a rate of 59.4%. In comparison, the PBS, **RGD-Dox**, and **TCO-P₄** groups did not exhibit significant tumor inhibition.

In our manuscript, we provide further clarification on this observation: “In contrast, mice administered an equivalent dose of **activable-Dox** experienced a significant inhibition of tumor progression, with a tumor growth inhibition value (TGI) was as high as 87.6% (Figure S70), underscoring the success of this PDC strategy *in vivo*.”

For additional details on these experiments, please refer to the Supplementary Information (Page S55), as outlined below:

17.2 Calculation of the tumor growth inhibition values (TGI)

Tumor growth inhibition values (TGI) were calculated using the following formula: $TGI (\%) = (1 - RTV_{\text{treatment}} / RTV_{\text{control}}) * 100$. RTV refers to relative tumor volume, representing the ratio of the volume at the end of treatment to the volume at the beginning of treatment.

Supplementary Fig. 70. TGI calculations for the four treatment groups. Error bars represent the standard deviation (n = 5).

Comments 5: The reference list can be improved. The #9 ref is not formatted in style. Some on-target bioorthogonal prodrug activation work should be cited, e.g. *Nat. Commun.*, 2018, 9, 5032; *Bioconjugate Chem.* 2023, doi: 10.1021/acs.bioconjchem.3c00404.

Response 5: We appreciate your valuable feedback. We have revised the format of reference #9 as follow:

9. Hennrich, U., Kopka, K. Lutathera®: The First FDA- and EMA-Approved Radiopharmaceutical for Peptide Receptor Radionuclide Therapy. *Pharmaceuticals* **12**, 114 (2019).

Furthermore, we have incorporated additional references in paragraph 3 of page 2 to enhance the citation of pertinent bioorthogonal prodrug activation studies:

“allow for specific bond breakage and release of payloads from the delivery machinery¹⁷⁻²⁶.”

25. Yao, Q., et al. Synergistic enzymatic and bioorthogonal reactions for selective prodrug activation in living systems. *Nat. Commun.* **9**, 5032 (2018).

26. Yao, Q., et al. A Dual-Mechanism Targeted Bioorthogonal Prodrug Therapy. *Bioconjugate Chem.* **34**, 2255–2262 (2023)

Comments 6: Several ¹H NMR spectra exhibit peaks around 1.6 ppm which were not attributed (e.g., compound 1). It is suggested that the author provides a proper description of the substance associated with these peaks.

Response 6: Thank you for your valuable feedback. Compound 1 has undergone re-purification, and we have

identified the H₂O peak around 1.6 ppm in relation to other compounds. Comprehensive details can be found in the Supplementary Information (Pages S57–S83), outlined below:

8.18
8.18
8.16
8.16
7.60
7.58
7.58
7.23
7.21
7.18
7.12
7.12
7.07
7.07
6.17
6.16
6.14
6.14
5.86
5.77
5.73
5.32
5.28
5.24

3.77
3.18
3.16
3.14
3.12
1.95
1.93
1.91
1.90
1.89
1.89
1.88
1.87
1.85
1.83
1.40
1.38
1.36
1.05
1.03
1.01

7.37
7.37
7.36
7.35
7.34
7.34
7.33
7.33
7.32
7.31
7.30
7.30
7.29
7.29

5.71

4.43

4.41

4.25

4.24

4.22

3.57

3.56

3.54

0.80

0.01

Your assistance in refining the manuscript is greatly appreciated.

Response to Reviewer 3:

This contribution by He et al describes an elegant prodrug design strategy that takes advantage of a drug release mechanism triggered by bioorthogonal conjugation. This release mechanism was first reported by Robillard in 2013 (ref 18) and then optimized by Robillard in 2020 (ref 20) and the authors of this contribution in 20212 (ref 21). The current submission applies this drug release mechanism to designing peptide -drug conjugates (PDCs), which have shown promise as privileged therapeutics. Specifically, the authors report that vinyl substituted tetrazines show facile conjugation to cysteine thiols., while maintaining their bioorthogonal conjugation with TCOs for drug release (Figure 2). The drug release mechanism has been realized on cultured cells (Figure 3-4) as well as in a mouse model of cancer (Figure 5). Particularly noteworthy is that the prodrugs (peptide-drug conjugates) affords cancer cell target capabilities, and upon activation (drug release) via TCO conjugation, kills cancer cells more efficiently than the corresponding drug-only controls. This is a creative study with carefully designed experiments to support the authors' conclusions. The significance of this work lies in the peptide-drug conjugation methodology (via vinyl tetrazine) as well as the in vivo demonstration of the power of the click-and-release strategy for designing peptide-drug conjugates as prodrugs. This work would be suitable for publication in Nature Comm after the following issues are addressed:

Comments 1: The observation of nanostructure formation (Figure 2) is interesting. However, the origin of the observed stability is not fully clear. Has the author examined the stability of the peptide-drug conjugate at below its critical micelle concentrations? This experiment will teach if the observed stability is intrinsic of the peptide-drug conjugates or due to micelle formation.

Response 1: We appreciate your comment. We conducted stability tests on **RGD-Dox** and **RGD-SN-38** in 10 mM GSH at a concentration of 2 μ M, which is below the critical micelle concentrations. After incubating at 37 $^{\circ}$ C for 12 hours, we observed that 95% of **RGD-Dox** and 85% of **RGD-SN-38** remained intact. This stability is slightly less than that observed at a 50 μ M concentration (96% for **RGD-Dox** and 92% for **RGD-SN-38**). These findings suggest that the observed stability is intrinsic to the peptide-drug conjugates, with a slight enhancement due to micelle formation.

These results have been explicitly incorporated into the manuscript, where we state:

Similarly, we found that 95% of **RGD-Dox** and 85% of **RGD-SN-38** remained intact at 2 μ M, below the critical micelle concentrations, after 12 hours of incubation at 37 $^{\circ}$ C (Figure S46). This indicates that the observed stability is intrinsic to the PDCs, with a slight enhancement by micelle formation.

Details of these experiments have been included in the Supplementary Information (Page S42), as outlined below:

10.2 Stability of peptide-conjugates in the presence of a large excess of GSH at 2 μ M

RGD-Dox and **RGD-SN-38** were diluted to a concentration of 2 μ M using PBS (10% DMSO, pH 7.4). A stock solution of GSH (400 mM in PBS) was then added to achieve a final concentration of 10 mM. The mixture was homogenized and incubated at 37 $^{\circ}$ C for 12 hours. Stability was assessed by measuring the decrease in peak area at 520 nm (with the initial peak area at 0 min defined as 100%). This measurement was monitored by HPLC-MS. Statistical analysis was performed on data from three replicates to calculate the mean and standard deviation.

Time (h)	Stability (%)	
	RGD-Dox	RGD-SN-38
0	100	100
1	97 ± 1	94 ± 2
3	94 ± 1	88 ± 4
5	95 ± 0.1	87 ± 6
8	95 ± 0.5	85 ± 7
12	95 ± 1	85 ± 3

Supplementary Fig. 46. Stability of RGD-Dox and RGD-SN-38 in the presence of a large excess of GSH at concentrations below their critical micelle concentrations.

Comments 2: The drug release appears to exhibit a two-phase mechanism (Figure 2F). A fast phase of 35 min and then a slow phase of 24 hrs. How should one understand and rationalize this two-phase behavior?

Response 2: We appreciate your helpful comment. The IEDDA reaction between *trans*-cyclooctene and tetrazine yields 4,5-dihydropyridazines (4,5-DHP), which can tautomerize into 1,4- and 2,5-dihydropyridazine (1,4- and 2,5-DHP) isomers. In a previous study on the release mechanism (*J. Am. Chem. Soc.* **142**, 10955-10963 (2020)), it was reported that 4,5-DHP predominantly tautomerizes to 2,5-DHP. Additionally, the study found that 2,5-DHP serves as the rapid releasing species, inducing a fast elimination, resulting in the initial fast phase of the release curve. Subsequently, the slow tautomerization of 1,4-DHP to 2,5-DHP leads to the second slow phase of the release curve. We speculate that a similar two-phase releasing behavior exists in our bioorthogonal cleavage reaction, involving the same 1,4-DHP and 2,5-DHP intermediates.

Your assistance in refining the manuscript is greatly appreciated.

REVIEWERS' COMMENTS

Reviewer #1 (Remarks to the Author):

The authors have addressed all the comments. I recommend the manuscript for publication. Please address the following minor comments.

Comments

Page 3, line 65: “with Cystic thiol” could be replaced with “with Cys thiol”.

Pages 18-19: Please check and correct references 34 and 35. Reference 34 should be Soudy, R., Mol. Pharmaceutics 14, 593–604 (2017) while reference 35 should be Su, H., J. Controlled Release 219, 383–395 (2015).

Reviewer #2 (Remarks to the Author):

The authors have addressed all the comments from the reviewers and the editor with additional studies and revisions. Therefore, I would suggest the acceptance of this manuscript in its current version.

Reviewer #3 (Remarks to the Author):

The authors have done a good job revising the manuscript and my initial concerns have been fully addressed.

Point-by-point responses to the reviewer comments:

Reviewer 1: The authors have addressed all the comments. I recommend the manuscript for publication. Please address the following minor comments.

Comment 1: Page 3, line 65: “with Cystic thiol” could be replaced with “with Cys thiol”.

Response 1: We appreciate your comment. We have corrected this error in the manuscript: “with Cys thiol”.

Comment 2: Pages 18-19: Please check and correct references 34 and 35. Reference 34 should be Soudy, R., Mol. Pharmaceutics 14, 593–604 (2017) while reference 35 should be Su, H., J. Controlled Release 219, 383–395 (2015).

Response 2: Thank you for your valuable feedback. We have revised the correct references in the manuscript: 34. Soudy, R., Etayash, H., Bahadorani, K., Lavasanifar, A., Kaur, K. Breast Cancer Targeting Peptide Binds Keratin 1: A New Molecular Marker for Targeted Drug Delivery to Breast Cancer. Mol. Pharmaceutics 14, 593–604 (2017). 35. Su, H., Koo, J. M., Cui, H. One-component nanomedicine. J. Controlled Release 219, 383–395 (2015).

Your assistance in refining the manuscript is greatly appreciated.

No comments were received from reviewers 2 and 3.